# Monitoring Behavior and Welfare of Cattle in Response to Summer Weather in an Arizona Rangeland Pasture Using a Commercial Rumen Bolus

**DOI:** 10.3390/ani15101448

**Published:** 2025-05-16

**Authors:** Amadeus O. Barto, Derek W. Bailey, Ly Ly Trieu, Pippa Pryor, Kieren D. McCosker, Santigo Utsumi

**Affiliations:** 1Animal & Range Sciences, New Mexico State University, Las Cruces, NM 88003, USA; amadeus1@nmsu.edu (A.O.B.); sutsumi@nmsu.edu (S.U.); 2Deep Well Ranch, Prescott, AZ 86304, USA; 3Computer Sciences, New Mexico State University, Las Cruces, NM 88003, USA; lytrieu@nmsu.edu; 4School of Agriculture and Food Sustainability, The University of Queensland, Gatton, QLD 4343, Australia; pippa.pryor@uq.edu.au; 5Centre of Animal Science, The University of Queensland, Gatton, QLD 4343, Australia; k.mccosker@uq.edu.au

**Keywords:** cattle behavior, precision livestock management, reticular temperature, activity, ambient temperature, relative humidity, wet bulb globe temperature, on-animal sensor, heat load, heat stress

## Abstract

Monitoring the welfare of cattle on rangelands is logistically challenging, and hot weather can adversely affect animal health and productivity. The aim of this study was to evaluate the use of a commercially available rumen bolus (SmaXtec) to measure the body temperature and behavior of cattle grazing rangelands during the summer in Arizona. The cattle body temperature increased with higher wet bulb globe temperatures, an index used to monitor heat stress in humans. Cattle activity levels were most closely associated with relative humidity, surprisingly increasing at more humid levels. The estimated water intake of cattle decreased at higher humidity levels. The rumen bolus appears to be an effective tool for remotely monitoring cattle activity and body temperature while grazing extensive pastures.

## 1. Introduction

Heat stress is experienced by cattle when they are exposed to high environmental temperatures and humidity for extended periods of time, exceeding the animals’ ability to regulate their body temperature effectively [1]. Rising temperatures, altered precipitation patterns and more frequent extreme weather events forecasted from climate change are expected to result in an increase in the number of livestock experiencing heat stress. These results are likely to impact the viability of outdoor livestock operations in coming years [2]. Climate change is expected to negatively affect rangeland grazing systems in sub-tropic and temperate zones [3,4].

Cattle experiencing a high heat load, exceeding their ability to regulate, will have a rise in core body temperature, which has been associated with behavioral changes resulting in modifications of water intake and feed intake and related to the activity level. These consequences of prolonged periods of heat stress have negative effects on production, reproduction and immune responses [5,6]. Historically, heat stress has had a substantial impact on the beef industry, with an annual loss of USD 369 million in the United States alone in 2003 [7]. Despite the current climate trends and the public’s concerns over animal welfare, little research has been conducted to understand the extent and consequences of heat stress for cattle grazing rangelands [8,9].

Rangelands are vast, often with rugged terrain, making observations of cattle welfare difficult and labor-intensive [10]. Recent advancements in modern technologies such as global positioning systems (GPS), accelerometers and Internet of Things (IOT) have facilitated the real-time remote monitoring of livestock on rangelands [11,12]. On-animal sensors have been shown to be useful in monitoring the health, behavior, activity levels and locations of livestock [13]. The SmaXtec bolus used in this study remotely monitors various physiological parameters, such as body temperature, and activity using sensors placed inside the reticula of cattle. A similar bolus using identifiable RFID has been applied in a feedlot situation as a non-intrusive tool to assess the heat loads of individual cattle [14]. Precision livestock management (PLM) is the integration of these modern technologies into the ranching industry, which creates opportunities to monitor and mitigate livestock welfare concerns in rangeland systems [15].

The objective of this case study was to evaluate the effectiveness of a commercial rumen temperature bolus in monitoring the behavior and welfare of cattle in response to summer weather conditions experienced on a Central Arizona rangeland.

## 2. Materials and Methods

### 2.1. Site and Environment

The study took place in the North Ditch Pasture (NDP) on Deep Well Ranch (DWR), located 16 km north of Prescott, Arizona, United States (34°42.8″ N, 112°25.8″ W). Deep Well Ranch comprises 8004 ha in total, with an elevation range of 1434 to 1675 m. The study pasture encompasses 312 ha, with an elevation gradient of 1460 to 1520 m. Deep Well Ranch falls within the cold semi-arid (Bsk) category of the Köppen climate classification, with average annual precipitation of 487 mm [16]. Evapotranspiration can sometimes exceed precipitation. The terrain of DWR is primarily rolling hills dominated by perennial grasses of black grama (*Bouteloua eriopoda* (Torr.)), dropseed (*Sporobolus* spp.) and purple three-awn (*Aristida purpurea* Nutt.).

### 2.2. Animals

A herd of 28 registered 2-year-old Corriente heifers was equipped with GPS ear tags to track movement. Ten of the 28 heifers were randomly selected and administered rumen temperature boluses. All heifers were kept together in the same pasture from 7 April to 4 September 2023, in conjunction with the ranch manager’s grazing system rotation plans. The study was conducted in accordance with the New Mexico State University Institutional Animal Use and Care Committee (protocol numbers 2303000200 and 2308000468).

### 2.3. Devices

#### 2.3.1. Global Positioning System and Accelerometer Ear Tags

All 28 heifers were tagged with 701x Autonomous Rancher xTPRO GPS tracking ear tags (Fargo, ND, USA), https://www.701x.com/xtpro, on 17 February 2023. The ear tags had an average weight of 46.15 g, including the male attachment pin. The tags were designed to record GPS locations at 15 min intervals and provide an activity index every 5 min. The plan was to use the tag data to help monitor cattle behavior and compare cattle activity levels from the boluses and the tags. However, most tags failed or fell off within 2 months of installation. Thus, all 701x tag tracking data were excluded from analysis.

#### 2.3.2. Rumen Temperature Bolus

SmaXtec Classic Boluses (Graz, Austria), https://smaxtec.com/en/smaxtec-system-in-detail/#boli (accessed 4 January 2025), were placed in the 10 randomly selected heifers on 7 April 2023. The boluses recorded 6 metrics: the reticular temperature (RT), adjusted reticular temperature (ART), activity index, rumination index, water intake index and estrus index. These metrics were measured every 10 s then averaged within 10 min intervals, except water intake, which was summarized to a single 24 h index value. The adjusted rumen temperature is a measure of the reticular temperature excluding temperature changes from drinking events. The boluses were equipped with accelerometer sensors to calculate activity and rumination indices; however, the raw data were not accessible.

SmaXtec boluses use long-range Bluetooth to communicate with a base station, which sends data in real time to the internet using a SIM card with a cellular network. To ensure consistent data collection and transmission, the base station was placed 40 m from the only water source within the study pasture. If a heifer does not visit the water source or spend adequate time within range of the base station, the bolus can store up to 6 days of data onboard. If the bolus does not offload the stored data to the base station within 6 days, the oldest stored values will be overwritten.

### 2.4. Weather Data

The weather variables used in this study were a combination of meteorological metrics and thermal indices.

#### 2.4.1. Weather Station

All weather data were collected at the Prescott Regional Airport (PRA), also known as Love Field (KPRC), in Prescott, Arizona, United States, with coordinates (34°38′57.0114″ N, 112°25′19.9914″ W), at an elevation of 1537 m. The weather station was 6.66 km from the water tank in the North Ditch study pasture. Weather metrics were accessed and downloaded as an Excel spreadsheet through MesoWest, https://mesowest.utah.edu/html/help/userguide.html (accessed 1 August 2023). The downloaded weather data consisted of the ambient temperature (AT), relative humidity (RH), wind speed and solar load at 5 min intervals. Precipitation data were not included in the analyses because rainfall occurred periodically, with most days having no measurable rainfall. The few rainfall events could not be adequately evaluated using our statistical analyses.

#### 2.4.2. Thermal Indices

Two thermal indices were used to evaluate the combined effect of summer weather variables. The first thermal index used was the temperature–humidity index (THI), calculated from the equations in [17]. The second thermal index used was the wet bulb globe temperature (WBGT). Localized WBGT values for the study site were accessed through https://klimoinsights.com/about-us/ (accessed 13 August 2024), created by Jordan Clark [18]. The WBGT index, RH and AT received from Klimo Insights were validated in August 2024 using data collected from a Kestrel 5400AG Cattle Heat Stress Tracker (Kestrel, Boothwyn, PA, USA), https://kestrelmeters.com/kestrel-5400ag-cattle-heat-stress-tracker-link-vane-mount?srsltid=AfmBOooavDINmMtS2Rds2vc9ZZaTJ06ixazHeLmsnUDWLLvIJq4X_dzF (accessed 4 January 2025). Pearson correlations were calculated between the measured (Kestrel) and reported (Klimo Insights) values for the WBGT, RH and AT. Simple regression was performed between the reported and measured values of WBGT. The WBGT data from Klimo Insights were used as the independent variable, and the Kestrel data were used as the dependent variable. These analyses were used to compare the weather data from the airport and Klimo Insights to the values recorded at the study site.

Bolus and weather data from 1 June to 29 August 2023 were analyzed for this study. The ambient temperature during this period varied between 7.0 and 24.1 °C, and the RH and WBGT varied from 2.0 to 100.0% and 1.9 to 30.8 °C, respectively (Table 1).

### 2.5. Cattle Observation

In-person, visual observations of cattle behavior were conducted between 6 June to 1 July 2023 and again from 10 August to 12 August 2023. Visual observations were used to train machine learning algorithms to classify behavior types using the activity and rumination indices of the SmaXtec boluses. All observations were made by one trained person between dawn and dusk, with the aid of binoculars and a spotting scope to identify cattle based on ear tag numbers. A total of 15,513 min of cattle observations were collected. Behavioral types were recorded in one-minute events. The behavioral types recorded were grazing, walking, standing, laying, drinking and ruminating. Behavioral types were categorized and recorded if the behavior spanned the length of a one-minute event, with any deviation in behavior lasting less than 30 s. Behaviors lasting less than 30 s were to be considered micro-behaviors, which were defined as a small burst of activity or momentary change in the activity type of an individual cow. These short bursts of behavior that could occur within an event-based activity type include scratching, licking, stepping (shifting of body weight) and headbutting. The definitions for the event behaviors were based on [19] and are described as follows.

Grazing: The individual stood with their head in the down position or walked for less than a minute between head-down bouts. Walking: An individual was continuously moving forward with their head in the up position. Standing: The individual was stationary with legs extended. Lying: The individual’s side was in contact with the ground. Ruminating: The individual could be visually observed chewing cud, including regurgitation, chewing and swallowing. Drinking: The individual was at the water with its head in the down position.

### 2.6. Annotation and Evaluation of Proprietary Bolus Indices

#### 2.6.1. Data

The 1 min observational data were matched to the 10 min SmaXtec data using the date and time. This was achieved by cross-referencing the observational data with the SmaXtec data, ensuring that the timestamp of the observational data was less than or equal to the timestamp of the SmaXtec event entry. Each 1 min interval of observation data was equivalent to 10% of one 10 min SmaXtec interval. These data were then summarized for each SmaXtec interval by calculating the percentage of time for which each behavior type was expressed within the event (termed “Timestamp”, 0–100%). For example, if three 1 min observations within a 10 min interval were grazing, the value would be 30% grazing. The same approach was applied to each behavior type defined in Section 2.5, displaying the percentage of each behavior type from 0 to 100% within each 10 min SmaXtec interval.

Two additional columns called “Inactive” and “Active” were created, to provide a simpler classification alternative. The “Inactive” column combined the percentages of the behavior types standing and lying. The “Active” column combined the percentages of grazing and walking. All SmaXtec data with a “Timestamp” less than 10% were excluded from any further evaluation. Rumination was evaluated independently (ruminating or not ruminating) using the same approach.

#### 2.6.2. Statistical Analysis

Three types of statistical analysis were performed to determine and assess the relationships between the observed behaviors and the SmaXtec activity index.

A Pearson correlation coefficient was calculated between the SmaXtec activity index and each behavior type from Section 2.5, excluding drinking. The timestamp value for each behavior type and associated SmaXtec activity index was used to identify the strength and direction of the relationship, whether positive or negative.

The second analysis performed was a linear regression using PROC MIXED in SAS [20]. The dependent variable was the behavior status (activity or rumination), and the model included the SmaXtec activity or rumination index as the fixed effect (simple linear regression using all paired data).

The third statical analysis performed was a repeated-measures analysis using PROC MIXED in SAS [20]. The covariant structure was autoregressive AR(1). The independent variable was the activity or rumination index, and the dependent variable was the activity type (active or inactive) or rumination, respectively. The heifer was the subject. The model evaluated linear, quadratic and cubic effects of the independent variables.

#### 2.6.3. Machine Learning

Decision trees, random forests and neural networks with 1 hidden layer were the machine learning algorithms used to test the classification accuracy of the activity and rumination indices compared to observation data, using the Waikato Environment for Knowledge Analysis (WEKA) version 3.8 software [21]. The classification structure was binary for the evaluation of both the activity and rumination indices. Ten-fold cross-validation was used in both the activity and rumination analyses. The observational data were separated into ten equal parts, with nine parts being used for training and one part for testing. The observational data used in the evaluation of the SmaXtec’s activity index were classified as active (combined walking and grazing) or inactive (lying and standing). The activity index was then used in the models to classify the remaining observational data as either active or inactive. The observational data used to evaluate the rumination index were ruminating or not ruminating and were independent of the activity type. The rumination index was then used in the models to classify the remaining observational data as ruminating or not ruminating. All models were run twice with and without cow in the model.

### 2.7. Data Organization of the Weather and Bolus Metrics

Both the SmaXtec and weather data were summarized within 3 h and 24 h intervals. This focused on the data of 1st June 2023 to 28 August 2023, with the exception of 12–14 July, as SmaXtec data transmission was interrupted due to ranch operations. The 3 h data were used to evaluate diurnal changes across the day (Table 2). Two data sets were compiled with both the SmaXtec and weather data: one for 24 h averages and another consisting of the eight corresponding 3 h time periods. The 24 h data consisted of all weather and SmaXtec metrics excluding the estrus and rumination indices. The 3 h data set contained all metrics excluding the SmaXtec’s estrus and water intake indices.

### 2.8. Statistical Analysis to Determine Relationship Between Weather and Bolus Metrics

#### 2.8.1. Three Hour

An independent repeated-measures analysis was performed for each combination of weather and SmaXtec metrics, excluding the water intake and estrus indices. The analyses were completed using PROC MIXED in SAS [20]. The covariate structure was autoregressive AR(1). The independent variable was one of the weather metrics (ambient temperature, °C; relative humidity; wind speed, m/s; solar load; WBGT; or THI). The dependent variable was one of the SmaXtec metrics (RT, ART, activity index or rumination index). Cow was the subject for all analyses. Linear, quadratic and cubic effects for each weather metric were evaluated. The best fitting models were selected based on the smallest Akaike Information Criterion (AIC) score [20].

#### 2.8.2. Twenty-Four Hour

An independent repeated-measures analysis was performed for each combination of weather and SmaXtec metrics, excluding rumination and estrus data. The analyses were performed using PROC MIXED in SAS [20]. The covariant structure was autoregressive AR(1). The independent variable was one of the weather metrics (ambient temperature, relative humidity, wind speed, solar load, WBGT or THI). The dependent variable was one of the SmaXtec metrics (RT, ART, activity index or water intake index). Cow was the subject for all analyses. Linear, quadratic and cubic effects were evaluated for each separate weather metric. The best-fitting models were determined based on the smallest AIC score [20].

## 3. Results

### 3.1. Proprietary Bolus Indices

#### 3.1.1. Correlations

A weak positive correlation of 0.159 was found between rumination recorded in cattle behavior observations and the rumination index. Additionally, there was a weak positive correlation of 0.067 between the activity index and walking observed during cattle behavior observations. However, a moderate positive correlation of 0.428 was observed between the activity index and grazing; furthermore, a moderate correlation of 0.44 was observed between the activity index and the active state (walking + grazing). A moderate negative correlation of −0.50 was calculated between the activity index and lying, compared to a weak negative correlation of −0.0865 between the activity index and standing.

#### 3.1.2. Simple Regression

Both the activity and rumination indices were found to significantly increase (*p* < 0.001) with increasing levels of observed activity and rumination (Table 3).

#### 3.1.3. Polynomial Regression with Repeated Measures

A significant relationship between the activity index and observed activity was found (*p* < 0.05) for all functions—linear, quadratic and cubic (Table 4). Changes in AIC scores were minimal, although the linear relationship was slightly favored (Table 4)

### 3.2. Machine Learning

Models that included cow had an increased classification rate of about 3% across algorithm types, regardless of which index was used. Neural networks with one hidden layer produced the highest correctly classified rate of 72.47% among all three algorithms using the activity index with heifer in the model. The lowest-preforming algorithm using the activity index was random forest, with a correct classification rate of 66.03%, with cow in the model.

The algorithm that correctly classified the most rumination events using the rumination index was a decision tree with a classification rate of 77.26%; comparatively, when cow was excluded from the model, the decision tree yielded a correct classification rate of 74.25%. Neural networks with one hidden layer had the lowest classification rate of the three algorithm types for the rumination index when heifer was included in the model, with a classification rate of 75.89%.

### 3.3. Relationships Between Weather and Bolus Metrics

#### 3.3.1. Weather Metrics

The weather metrics used as independent variables were checked for multicollinearity (Table 5). All weather metrics were positively correlated, except the RH, which was negatively correlated with all the other weather metrics. Most of the correlations were moderate. However, a strong correlation was found between the WBGT and solar load. Very strong correlations were found between the ambient temperature, THI and WBGT (Table 5). The moderate and strong correlations among the weather metrics may suggest multicollinearity.

The weather metrics (WBGT, AT, and RH) recorded by a Kestrel device were highly correlated (*p* < 0.01) with the values provided by Klimo Insights (Table 6). The linear regression of the Kestrel WBGT and the WBGT provided by Klimo Insights resulted in a slope of 0.65 +/− 0.05 SE, with a y-intercept of 4.13 +/− 1.15 SE. The R-squared value for this regression was 0.76.

#### 3.3.2. Three Hour

The bolus metrics averaged into 3 h periods revealed the RT and ART to nearly be the same in the early morning and late at night (Figure 1). A clear divergence between the RT and ART occurred later in the morning (06:00 to 08:00) and slowly recovered over the afternoon. The reticular temperature experienced a larger drop than the ART. The mean RT between 06:00 and 08:00 was 39.39 °C, with an SE of 0.013, compared to the ART, with a mean of 39.08 °C and an SE of 0.007 for the same period. However, the two followed a similar daily pattern. All RT periods were different (*p* < 0.01), except periods 03:00 to 05:00 and 15:00 to 17:00 (*p* = 0.71). Similarly, all ART periods were different (*p* < 0.01), except periods 00:00 to 02:00 and 15:00 to 17:00 (*p* = 0.1).

The repeated-measures analyses found that the best-fitting model for the ART was a cubic relationship with the WBGT (Table 7, Figure 2 and Figure 3). The second-best model in describing changes in the ART was a cubic relationship with the RH (Table 7).

The relationship between the ART and WBGT appeared to be inverse when graphed together (Figure 2). The cubic relationship indicates that the ART initially dropped as the WBGT increased. The adjusted reticular temperature then gradually leveled out and increased when the WBGT was greater than 15 (Figure 3). The cubic relationship between the ART and RH revealed that the ART rapidly increased until the RH reached 40%; then, the ART continuously declined until RH of 85%, at which the ART began to increase again.

The activity index shows a clear diurnal grazing pattern (Figure 4). The best-fitting model for activity was a cubic relationship between RH and the activity index (Table 7, and Figure 5). The solar load had a negative linear relationship with activity and was the second-best model in describing changes in the activity index (Table 7).

A high spike in activity levels occurred around dawn, mid-morning and late evening near dusk (Figure 4). The activity index also suggests that the cows moved much less at night, despite activity having a negative linear relationship with the solar load (Table 7). The modeled activity levels steadily increased as the RH increased up to 50%. If the RH was greater than 50%, any increase in the activity index was gradual (Figure 5).

Changes in the rumination index in response to weather were less apparent. Most of the models evaluating the relationship between the rumination index and weather metrics were not significant (*p* > 0.05, Appendix A). Furthermore, models with the rumination index and weather metrics consistently had the largest standard errors in relation to differences in the means, which could have been, in part, a consequence of the larger numerical values used in the rumination index.

The wind speed was the best weather metric (cubic relationship) in modeling changes in the rumination index (Table 7 and Figure 6). The modeled rumination index increased more rapidly when the wind speeds surpassed 6 m/s (Figure 7). The changes in the rumination index were relatively small throughout the day, despite the large numerical value of the index value. The standard error of the rumination index for each period was larger than the differences among the time periods (Figure 7). A quadratic relationship with the ambient temperature was the second-best model for the rumination index (Table 7). Rumination decreased as the ambient temperature increased to 25 °C, after which rumination increased.

#### 3.3.3. Twenty-Four Hour

The repeated-measures analyses of the 24 h data were generally consistent with the analyses of the 3 h data sets. The relative humidity had the best relationships with the activity index in both the 3 h and 24 h data sets (Table 8). The wet bulb globe temperature was the best weather metric in modeling changes in ART in both data types. The 24 h data showed the ART to be consistently higher than the RT over the course of the study, with smaller variations in values (Figure 8). The observed differences between the RT and ART values were similar to the differences found between these variables in the 3 h data. The AIC scores for the ART were consistently smaller than those of the other bolus metrics (Table 8 and Table 9). The same result was found in the 3 h data (Appendix A), and, for this reason, more focus was given to the ART than other bolus metrics.

The best-fitting model for the ART was a quadratic relationship with the WBGT (Table 8 and Table 9). The visual representation suggests that the ART decreases as the WBGT increases when the values are less than 16. However, once the WBGT values surpass 16, the ART increases (Figure 9). The THI was the second-best predictor of the ART, with a positive linear relationship (Table 8 and Table 9). As the THI increased, the ART increased.

The activity index was best modeled with RH using a cubic function (Table 9). The modeled visual representation suggests that the activity levels increase quickly when the RH rises above 65% (Figure 10). A quadratic relationship with the solar load and the activity index was the second-best in modeling the activity index (Table 9). Activity declined as the average daily solar load increased up to 325 watts, and, on days when the average solar load was greater than 325 watts, the activity increased.

Water intake was only available in the 24 h data set. The water intake index was best modeled with RH using a linear relationship (Table 8 and Table 9). The water intake index decreased linearly with increasing RH (Table 9 and Figure 11). Days with the highest water intake had the lowest relative humidity (Figure 12). The second-best weather metric in modeling changes in the water intake index was the solar load, with a linear relationship (Table 8 and Table 9). The water intake index increased with increasing solar loads (Table 9). Interestingly, an inverse relationship between the RT and the water intake index was observed over the course of the study (Figure 13). Days with higher water intake indices often had lower RT values.

## 4. Discussion

### 4.1. Proprietary Bolus Indices

Validating the proprietary SmaXtec bolus indices was difficult because they are 10 min averages of accelerometer data. Multiple behavior types often occur within a single 10 min interval, which is only represented by one activity index. If the boluses’ reporting intervals had been more frequent, the statistical analyses (correlation and regression) may have been more informative. Smaller reporting intervals are more likely to capture only one behavior type. However, the algorithms and associated reporting intervals used by the SmaXtec bolus are proprietary and cannot be changed by users. The 10 min length of the reporting interval is likely designed by the manufacture to extend the battery life and ensure that the data packets are not too large to transmit when the animal is near the reader. Machine learning analyses suggest that the SmaXtec activity and rumination indices have some potential to monitor changes in cattle’s welfare on rangelands. Accuracies of 70% and greater were found with some machine learning algorithms when the cow was included in the model. The correct classification rates of all algorithms declined when the cow was excluded from the model. Similarly, Chang et al. [22] showed that developing an algorithm for an individual animal produced more accurate results compared to algorithms developed using a group. Individual-based algorithms may be more accurate for several reasons. Tobin et al. [23] discuss how the variation in the intensity of movements such walking or grazing differs among individual animals, which could cause errors in group-based algorithms. They also found variations between accelerometer devices of the same brand and type, which could introduce further errors when using a group-based algorithm. The proprietary algorithms used to calculate the indices in the SmaXtec boluses are uniform across devices and are intended to adjust to the individual animal based on a 7-day average from previous index readings. The bolus algorithms were designed for application in dairy systems. Dairy cows are larger in size than the Corriente heifers used in this study. More importantly, dairies are intensive systems where high-quality feeds are fed to cows in pens as compared to a rangeland system, where cattle must graze to obtain forage [13,24]. Thus, it is not surprising that the proprietary algorithms developed by SmaXtec for dairy cows in intensive systems may be less accurate when used for grazing cattle on extensive rangelands. Algorithms to monitor the activity and rumination of rangeland cattle based on accelerometers in rumen boluses should be developed separately using beef cattle breeds on pasture for development and validation.

### 4.2. Relationship Between Weather and Bolus Metrics

The higher values of the ART compared to the RT during midday demonstrate that the proprietary algorithm used in the ART does account for at least some of the drops in the reticular temperature resulting from drinking water. The temperature of the drinking water is cooler than the RT; thus, consuming water lowers the RT. The values of the ART appear more consistent than those of the RT during periods when cattle normally drink. Thus, the ART should be a better metric than the RT when used as an estimate of the core body temperature. However, neither the ART nor RT fluctuated by more than 1 °C daily. As mentioned by Shepard et al. [1], the homeothermy of cattle can fluctuate by 1 °C daily and varies between species, seasons and lifestyle stages, which creates difficulty in establishing a “normal” range for the core body temperature. Therefore, the cattle in this study likely did not experience heat stress. More research is needed in an area with hot temperatures lasting for longer durations than in Central Arizona, as the cool nighttime temperatures at the study site would mitigate the heat load. However, higher ART values were observed during periods with a higher WBGT, which suggests that producers in the region should monitor the WBGT rather than just the ambient temperature or THI to identify periods with an increased risk of heat stress.

The reticular temperature shows potential to provide information about the frequency and duration of drinking events among rangeland cattle. Ammer et al. [25] reported that the magnitude of the decrease in RT was associated with increasing water intake. We found that drops in RT occurred when cattle were observed drinking. Alternatively, Bewley et al. [26] found that the water temperature had a greater influence over drops in RT than the amount of water consumed. Ultimately, more research is needed to understand and validate the relationship between the RT, water temperature and quantity of water consumed.

The relative humidity was related to many of the SmaXtec bolus metrics. The relative humidity was one of the top two weather metrics associated with changes in many of the SmaXtec metrics, in both the 24 h and 3 h data sets—specifically, water intake, RT and activity. We found the strong association of the RH and SmaXtec metrics to be surprising. Initially, we had expected the THI or WBGT (which are combinations of multiple meteorological factors) to be the overall most associated weather factor. We speculate that the reason that the RH is so influential is due to the association with monsoonal weather patterns. Periodic rains and cloudy weather occurred during the monsoon season (July and August). Days with higher RH are correlated with wetter days in the region [27]. Increases in low atmospheric clouds decrease radiant heating from the sun [28], which would alter the heat exchange of heifers with their environment [29]. The water intake index decreased when the RH levels increased throughout the study. Furthermore, over the course of the study, the heifers’ activity levels increased, mirroring the rises in RH. Monitoring changes in RH could help producers in the region to anticipate changes in a herd’s daily behavior.

The solar load was also associated with the activity index and RT. The relationship between the solar load and activity was expected. Cattle normally are less active at night (no solar load), with peaks during the early morning and evening. When the solar load is high, around mid-afternoon, cattle typically rest during the summer [30].

### 4.3. Detecting Diurnal Patterns with Bolus Metrics

The heifers’ activity indices and core body temperatures (ART) reflect the diurnal patterns observed in other cattle studies. The ART peaked between 18:00 and 20:00. The adjusted reticular temperature then decreased, with the lowest levels between 08:00 and 10:00. The trend in the ART resembled a diphasic pattern. Wren et al. [31] evaluated the diurnal body temperature patterns in dairy cows and found 67% of the study cows to demonstrate diphasic patterns. Lees et al. [14] reported a diurnal rhythm where the RT increased from 08:00 to 20:00 and then declined from 20:00 to 08:00. These findings are congruent with our results. Similarly, Simmons et al. [32] described a diurnal pattern in the cattle reticulum temperature, where the peak occurred in the late evening or early morning.

The daily activity of the heifers also varied diurnally, using a 3 h block, peaking twice. The first peak occurred between 06:00 and 08:00—the period following sunrise. The second peak occurred between 18:00 and 20:00—the period prior to and during sunset. A small, localized peak occurred at midday between 12:00 and 14:00. The peaks in the SmaXtec activity index were similar to the reported diurnal grazing patterns. Free-range ruminants naturally display diurnal grazing patterns with morning and evening grazing bouts. [30]. Diurnal grazing patterns can change with the seasons [33]. These authors found that, during late July to early August, cattle spent the most time grazing immediately after sunrise and immediately before sunset, with a spike in the middle of the afternoon. This pattern was also observed by Parsons et al. [34] during the summer in the mountains of Oregon. They found that cattle spent the most minutes per hour grazing between 05:00 and 08:00 and again between 18:00 and 20:00. Parsons et al.’s [34] data also show an afternoon spike between 12:00 and 14:00. The SmaXtec activity index shows some potential to remotely monitor the time spent grazing a in cattle on rangelands; however, we did not find strong correlations or statistical relationships between the activity index and observed cattle behavior. This is likely a result of the proprietary algorithms for the activity index being developed in production systems such as dairies and feedlots, which differ from rangeland environments. In rangeland systems, animals have greater flexibility to express individual behaviors throughout the day, rather than feeding at routine times, which may impact the algorithm’s effectiveness [10,35,36]. Additionally, the resolution may be insufficient to detect grazing behavior. Averaging three-axis accelerometer data over 10 min intervals into a single value rather than recording multiple axes over shorter intervals (e.g., 1-min intervals) is likely less accurate and effective. More research, and perhaps alternative algorithm development, is needed to improve the accuracy of the SmaXtec indices for cattle grazing rangelands.

## 5. Conclusions

The commercial bolus showed promise in assessing cattle welfare during the summer months, with its metrics successfully used to describe the relationships between behavioral and physiological changes and changes in weather factors. The wet bulb globe temperature and relative humidity were associated with the adjusted reticular temperature. The relative humidity appears to be a key weather variable affecting heifers’ daily activity patterns and water intake. Monitoring the relative humidity in the southwest region during the summer could be useful for ranchers to indirectly estimate changes in cattle behavior. The wet bulb globe temperature may be a useful metric to identify periods when cattle may be subject to heat stress. The diurnal patterns observed in the adjusted reticular temperature and the activity index match the reported diurnal patterns in body temperature and grazing. The relationships between the adjusted reticular temperature and wet bulb globe temperature, as well as the relative humidity, in this study suggest that a commercial rumen bolus can be a useful tool for the monitoring of cattle welfare on rangelands during the summer. Further research with different breeds and rangeland locations is needed to validate the bolus’ value as a remote on-animal monitoring tool for cattle grazing rangelands.

## Figures and Tables

**Figure 1 animals-15-01448-f001:**
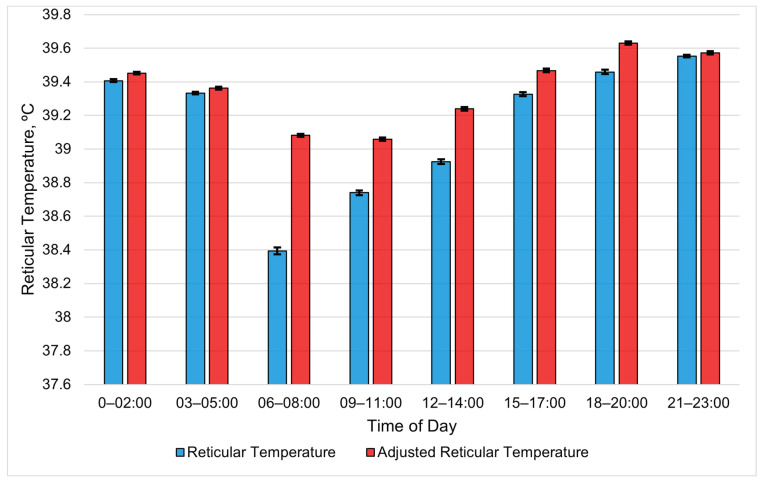
Least-square means of reticular temperature (RT) and adjusted reticular temperature (ART) averaged for each 3 h period during the summer study period. All RT periods are different (*p* < 0.01), except periods 03:00 to 05:00 and 15:00 to 17:00 (*p* = 0.71). Similarly, all ART periods are different (*p* < 0.01), except periods 00:00 to 02:00 and 15:00 to 17:00 (*p* = 0.1). Error bars represent standard errors.

**Figure 2 animals-15-01448-f002:**
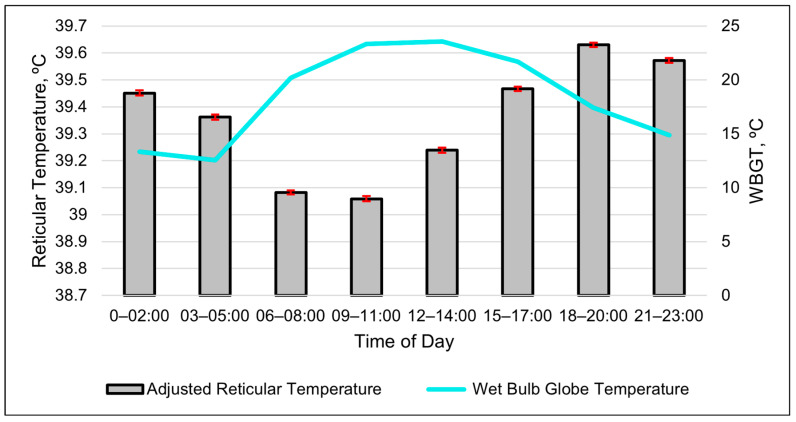
Relationship between the adjusted reticular temperature and wet bulb globe temperature (WBGT) over the course of a day using 3 h data. Columns are the least-square means of the adjusted reticular temperature, and error bars represent standard errors. Values are averages of the entire summer study period.

**Figure 3 animals-15-01448-f003:**
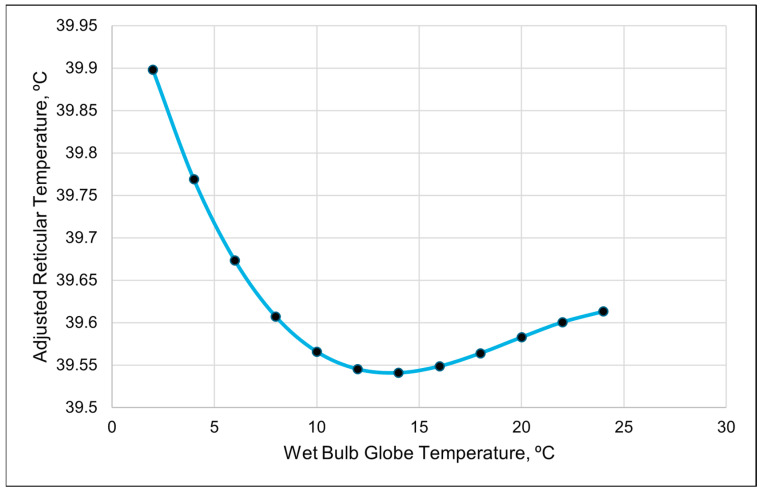
A visual representation of the cubic relationship between the adjusted reticular temperature and wet bulb globe temperature using 3 h data. This line was calculated from the coefficients obtained from the repeated-measures analysis (Table 7). The values on the *x*-axis were chosen to better illustrate the shape of the relationship and may exceed those observed in the study.

**Figure 4 animals-15-01448-f004:**
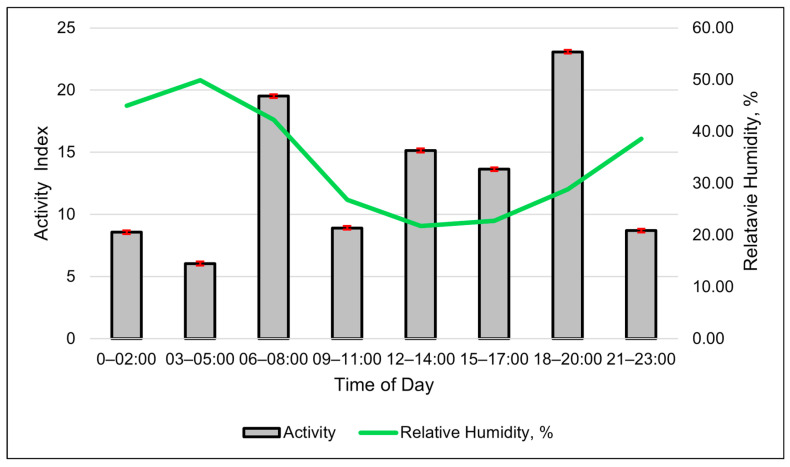
Relationship between the SmaXtec activity index and relative humidity throughout the day during using 3 h data. Columns are the least-square means of the activity index, and the error bars represent standard errors. Values are averages of the entire summer study period.

**Figure 5 animals-15-01448-f005:**
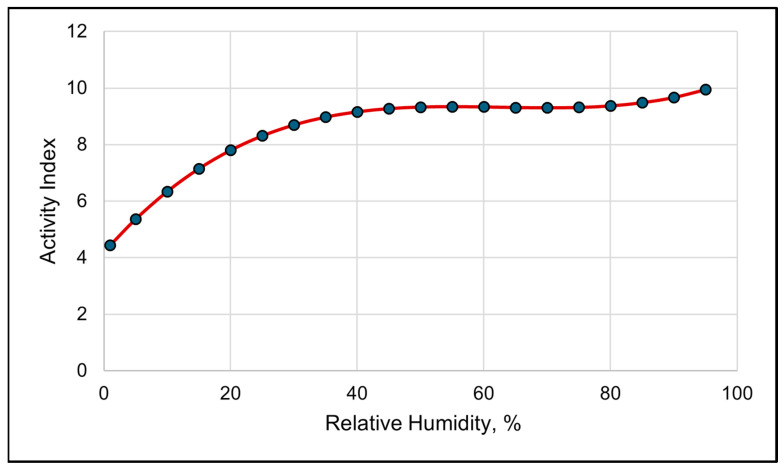
A visual representation of the cubic relationship between the activity index and relative humidity using 3 h data. This line was calculated from the coefficients obtained from the repeated-measures analysis (Table 7). The values on the *x*-axis are given to better illustrate the shape of the relationship and may exceed those observed in the study.

**Figure 6 animals-15-01448-f006:**
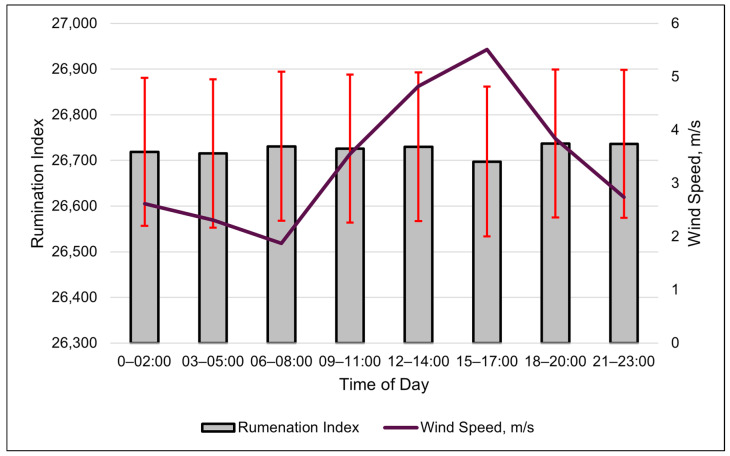
Relationship between the SmaXtec rumination index and wind speed (m/s) over the course of a day, using 3 h data. Columns are the least-square means of the rumination index, and the error bars represent standard errors. Values are averages of the entire summer study period. No differences in rumination index were detected (*p* > 0.05) among time periods.

**Figure 7 animals-15-01448-f007:**
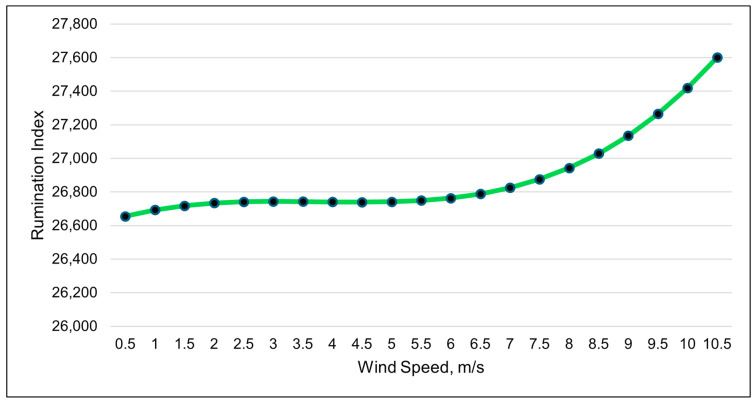
A visual representation of the cubic relationship between the rumination index and wind speed (m/s) using 3 h data. This line was calculated from the coefficients obtained from the repeated-measures analysis (Table 7). The values on the *x*-axis are given to better illustrate the shape of the relationship and may exceed those observed in the study.

**Figure 8 animals-15-01448-f008:**
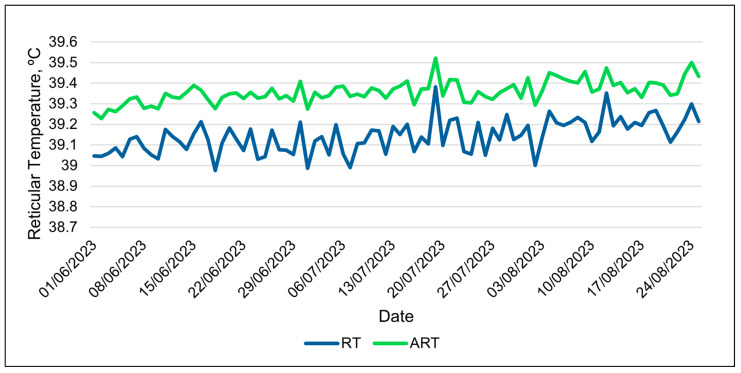
Daily changes in adjusted reticular temperature (ART) and reticular temperature (RT) during the entire summer study using 24 h data.

**Figure 9 animals-15-01448-f009:**
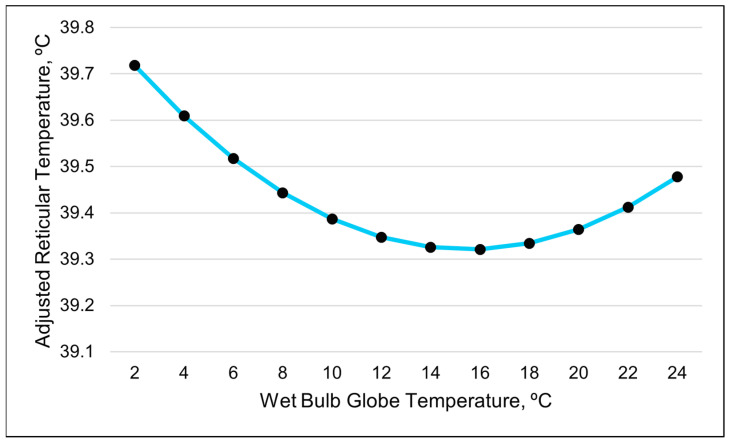
A visual representation of the quadratic relationship between the adjusted reticular temperature and the wet bulb globe temperature (WBGT), using 24 h data (daily averages). This line was calculated from the coefficients obtained from the repeated-measures analysis. The values on the *x*-axis are included to better illustrate the shape of the relationship and may exceed those observed in the study.

**Figure 10 animals-15-01448-f010:**
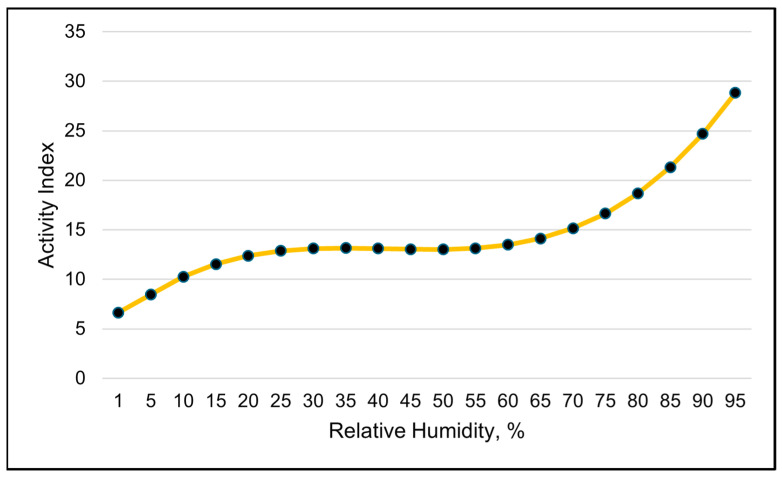
A visual representation of the cubic relationship between the activity index and relative humidity, using 24 h data (daily averages). This line was calculated from the coefficients obtained from the repeated-measures analysis. The values on the *x*-axis are used to better illustrate the shape of the relationship and may exceed those observed in the study.

**Figure 11 animals-15-01448-f011:**
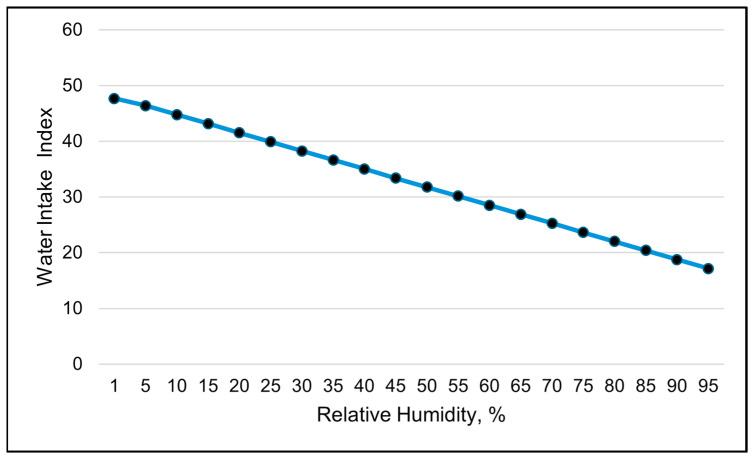
A visual representation of the linear relationship between the water intake index and relative humidity, using 24 h data (daily averages). This line was calculated from the coefficients obtained from the repeated-measures analysis. The values on the *x*-axis are used to better illustrate the shape of the relationship and may exceed those observed in the study.

**Figure 12 animals-15-01448-f012:**
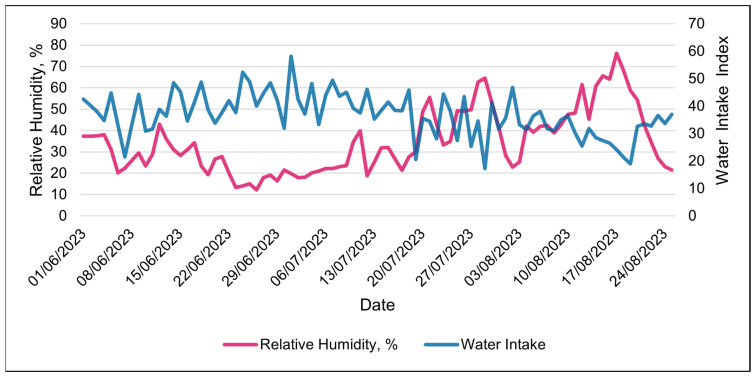
Daily relationship between SmaXtec’s water intake index and relative humidity during the entire summer study using 24 h data.

**Figure 13 animals-15-01448-f013:**
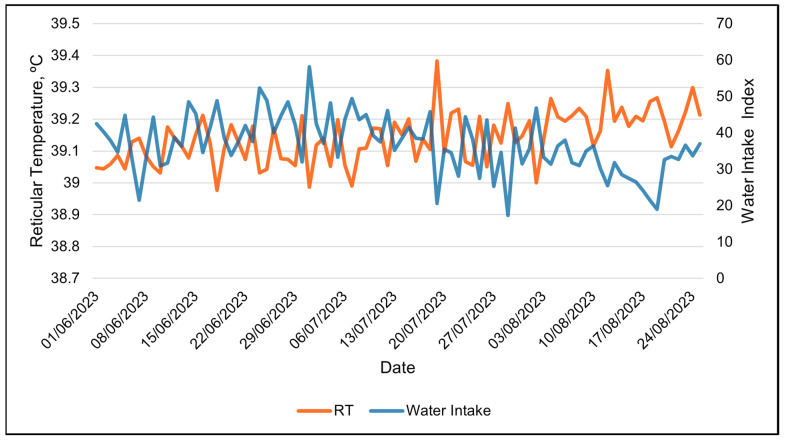
Daily relationship between reticular temperature and water intake index during the entire summer study using 24 h data.

**Table 1 animals-15-01448-t001:** Maximum, minimum and mean weather metric values based on 5 min weather data during the summer study period for the weather metrics used in the statistical analyses.

Metric ^1^	AT (°C)	RH (%)	Wind Speed (m/s)	THI (°C)	WBGT (°C)	Solar Load (Watts)
Mean	24.11	34.31	3.40	20.17	18.72	301.37
Max	39.40	100.00	14.40	27.17	30.83	938.00
Min	7.00	1.96	0.00	8.00	1.94	0.00

^1^ AT—ambient temperature, RH—relative humidity, THI—temperature–humidity index, and WGBT—wet bulb globe temperature.

**Table 2 animals-15-01448-t002:** Description of time periods used in the 3 h analyses (hours of each period).

Time Period	Hour
0	00:00–02:59
1	03:00–05:59
2	06:00–08:59
3	09:00–11:59
4	12:00–14:59
5	15:00–17:59
6	18:00–20:59
7	21:00–23:59

**Table 3 animals-15-01448-t003:** Coefficients, standard errors, R-squared values, *p*-values and Akaike Information Criterion (AIC) scores from simple regression between the observed activity and the activity index and regression between the observed rumination and the rumination index.

	Coefficient	SE	R^2^	*p*-Value	AIC
Activity	2.821	0.172	0.27	<0.0001	7507.3
Rumination	9.343	2.150	0.03	<0.0001	7466.4

**Table 4 animals-15-01448-t004:** Coefficients, standard errors, *p*-values and Akaike Information Criterion (AIC) scores for each function from the polynomial regression with repeated measures between the observed activity and the activity index with linear, quadratic and cubic relationships.

Polynomial Regression: Activity	
Linear	
Coefficient X	2.05
SE	0.189
*p*-value	<0.0001
AIC	7181.5
Quadratic	
Coefficient X	3.524
SE	0.656
*p*-value	<0.0001
Coefficient X2	−0.0481
SE	0.021
*p*-value	<0.05
AIC	7182
Cubic	
Coefficient X	8.366
SE	1.707
*p*-value	<0.0001
Coefficient X2	−0.391
SE	0.113
*p*-value	<0.01
Coefficient X3	0.007
SE	0.002
*p*-value	<0.01
AIC	7183.5

**Table 5 animals-15-01448-t005:** Correlation coefficients among weather metrics used in the statistical analyses. Weather metrics include ambient temperature (AT), relative humidity (RH), wind speed, temperature–humidity index (THI), wet bulb globe temperature (WBGT) and solar load.

	AT	RH	Wind Speed	THI	WBGT	Solar Load
AT	1					
RH	−0.515	1				
Wind Speed	0.446	−0.469	1			
THI	0.972	−0.344	0.368	1		
WBGT	0.848	−0.163	0.198	0.897	1	
Solar Load	0.566	−0.436	0.347	0.509	0.705	1

**Table 6 animals-15-01448-t006:** Pearson coefficient correlation results between the weather metrics (wet bulb globe temperature (WBGT), relative humidity (RH) and ambient temperature (AT)) provided from Klimo Insight using weather data from the Prescott Regional Airport and a Kestrel 5400AG cattle heat load tracker.

Weather Metric	Correlation
AT	0.93
RH	0.92
WBGT	0.87

**Table 7 animals-15-01448-t007:** The two best models describing the relationships between each SmaXtec metric (activity index, reticular temperature (RT), adjusted reticular temperature (ART) and rumination index) and the corresponding weather variables (ambient temperature (AT), relative humidity (RH), wind speed, wet bulb globe temperature (WBGT) and solar load) determined by the lowest Akaike Information Criterion (AIC) score using the 3 h data. The model type and corresponding standard errors and *p*-values are given for each coefficient in each model.

Bolus Metric	Weather Variable	Model Type	Function(s)	Coefficient	SE	*p*-Value	Model AIC
Activity	RH	Cubic	Linear	25.4078	3.0790	<0.0001	40,095.2
			Quadratic	−41.2614	7.4537	<0.0001	
			Cubic	21.9897	5.3174	<0.0001	
Activity	Solar Load	Linear	Linear	−0.0107	0.0007	<0.0001	40,110.3
RT	Solar Load	Linear	Linear	−0.0007	0.0001	<0.0001	5941.8
RT	RH	Linear	Linear	0.2519	0.0289	<0.0001	5981.9
ART	WBGT	Cubic	Linear	−0.0938	0.0111	<0.0001	−3876.6
			Quadratic	0.0053	0.0007	<0.0001	
			Cubic	−0.0001	0.0000	<0.0001	
ART	RH	Cubic	Linear	1.0712	0.1730	<0.0001	−3851.2
			Quadratic	−2.0390	0.3985	<0.0001	
			Cubic	1.1051	0.2795	<0.0001	
Rumination	Wind Speed	Cubic	Linear	120.1400	61.7424	0.0517	116,545
			Quadratic	−33.1055	14.8805	0.0261	
			Cubic	2.9257	1.0748	0.0065	
Rumination	AT	Quadratic	Linear	−71.8109	23.4900	0.0022	116,556.2
			Quadratic	1.4327	0.4488	0.0014	

**Table 8 animals-15-01448-t008:** The top two weather metrics, including the relative humidity (RH), solar load, wet bulb globe temperature (WBGT), wind speed and ambient temperature (AT), associated with changes in the SmaXtec indices, which included the activity index, reticular temperature (RT), adjusted reticular temperature (ART), rumination index and water intake index, from both the 3 h and 24 h data sets. The table provides the model rank, model type and Akaike Information Criterion (AIC) score for each model.

Data Type	SmaXtec Metric	Model Rank	Weather Metric	Model Type	AIC
3-h	Activity Index	1	RH	Cubic	40,095.2
3-h	Activity Index	2	Solar Load	Linear	40,110.3
24-h	Activity Index	1	RH	Cubic	3202.1
24-h	Activity Index	2	Solar Load	Quadratic	3270
3-h	RT	1	Solar Load	Linear	5941.8
3-h	RT	2	RH	Linear	5981.9
24-h	RT	1	RH	Linear	−961.1
24-h	RT	2	WBGT	Linear	−946.6
3-h	ART	1	WBGT	Cubic	−3876.6
3-h	ART	2	RH	Cubic	−3851.2
24-h	ART	1	WBGT	Quadratic	−1499.2
24-h	ART	2	THI	Linear	−1498.4
3-h	Rumination Index	1	Wind	Cubic	116,545
3-h	Rumination Index	2	AT	Quadratic	116,556.2
24-h	Water Intake	1	RH	Linear	6326
24-h	Water Intake	2	Solar Load	Linear	6392.1

**Table 9 animals-15-01448-t009:** The two best models determined by the lowest Akaike Information Criterion (AIC) score in describing the relationships between SmaXtec metrics, including the activity index, reticular temperature (RT), adjusted reticular temperature (ART) and rumination index, and corresponding weather variables, which included the relative humidity (RH), wet bulb globe temperature (WBGT), temperature–humidity index (THI) and solar load, using the 24 h data. Model types, coefficients, standard errors and *p*-values are given for each coefficient in each model.

Bolus Metric	Weather Variable	Model Type	Function(s)	Coefficient	SE	*p*-Value	Model AIC
Activity	RH	Cubic	Linear	53.8130	7.3369	<0.0001	3202.1
			Quadratic	−133.8700	18.5300	<0.0001	
			Cubic	107.7900	14.4182	<0.0001	
Activity	Solar Load	Quadratic	Linear	−0.0618	0.0086	<0.0001	3270
			Quadratic	0.0001	0.0000	<0.0001	
RT	RH	Linear	Linear	0.2403	0.0484	<0.0001	−961.1
RT	WBGT	Linear	Linear	0.0101	0.0025	<0.0001	−946.6
ART	WBGT	Quadratic	Linear	−0.0675	0.0211	0.0014	−1499.2
			Quadratic	0.0022	0.0006	0.0002	
ART	THI	Linear	Linear	0.0073	0.0019	0.0002	−1498.4
Water Intake	RH	Linear	Linear	−32.5000	2.1697	<0.0001	6326
Water Intake	Solar Load	Linear	Linear	0.0907	0.0075	<0.0001	6392.1

## Data Availability

Dataset available on request from the authors.

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
