# Peer review of "Monitoring Behavior and Welfare of Cattle in Response to Summer Weather in an Arizona Rangeland Pasture Using a Commercial Rumen Bolus"

_animals, 2025, doi:10.3390/ani15101448_

Round 1

Reviewer 1 Report

Comments and Suggestions for Authors

Dear authors and editor, I find the work interesting and well-presented. I want to send some general comments:
Line 1: Research paper only
Lines 5 and 6: Email addresses should be included in the affiliations, not next to the author's name
Lines 86-87: Scientific names should be italicized
Lines 168 to 175: The ethogram should be supported by bibliographic references
Lines 514 to 516: "More importantly, dairies are intensive systems where high-quality feeds are fed to cows in pens rather than a rangeland system where cattle must graze to obtain forage." Bibliographic references must support this sentence
Lines 590-592: "In rangeland systems, animals have greater flexibility to express individual behaviors throughout the day rather than feeding at routine times." Bibliographic references must support this sentence

Author Response

Our responses to Reviewer 1 comments are best viewed in the Word document chosen, but are listed below.

Line 1: Research paper only

Changed to Article based on Animals journal template

Lines 5 and 6: Email addresses should be included in the affiliations, not next to the author's name

For the Animals journal, email addresses are listed next to the name in published version.  Correspondingly, we placed the emails next to the Author’s names.

Lines 86-87: Scientific names should be italicized

Completed

Lines 168 to 175: The ethogram should be supported by bibliographic references

Added reference

Lines 514 to 516: "More importantly, dairies are intensive systems where high-quality feeds are fed to cows in pens rather than a rangeland system where cattle must graze to obtain forage." Bibliographic references must support this sentence

References added

Lines 590-592: "In rangeland systems, animals have greater flexibility to express individual behaviors throughout the day rather than feeding at routine times." Bibliographic references must support this sentence

References added

Reviewer 2 Report

Comments and Suggestions for Authors

Dear authors,

Sugestion of title: Behavior and Well-being of Cattle in Response to Summer Weather in an Arizona Rangeland Pasture

Line 16 to 18: This sentence is not clear

Line 23: remove this sentence

Line 34 to 35: It's necessary to include this sentence?

Line 35 to 39: The results should be described with values ​​and in a clearer way. Here they are written in a broad way and do not attract the reader's attention. Redoing is necessary.

Introduction: The objective of this case study was to evaluate the... or The objective of this research was to evaluate? It´s diference  in abstract and introduction. 

Line 91 to 92: It's 7 to September 4, 2023 and in abstract June 1 to August 29, 2023. Wich?

Line 130:  "Precipitation data were not included in the analyses because rainfall occurred periodically with most days having no measurable rainfall. " So precipitation doesn't interfere with the animal's confort?

Table 1: Is it necessary to insert this table in the text? This is data that you used for statistics. Describe in the text the average values ​​of each variable within the period studied, with standard deviation.

Line 154 to 175: How many people observed the behaviors? Did you use any reference methodology? Were the observers trained? Describe this information.

Table 2: Is it necessary to insert this table in the text?

Line 499 to 501: And why weren't they recorded at shorter intervals?

Line 599 to 601: This sentence is not necessary.

Author Response

Our responses to Reviewer 2 comments are best viewed in the attached Word document, but they are also listed below.

Sugestion of title: Behavior and Well-being of Cattle in Response to Summer Weather in an Arizona Rangeland Pasture

We appreciate the suggestion to shorten the title, however the use of a commercial bolus is the most practical and probably the biggest scientific value of this paper.  We think that including commercial rumen bolus in the title is useful and will help the reader decide if they want to read the paper. 

Line 16 to 18: This sentence is not clear

Sentence was revised and should be clearer.

Line 23: remove this sentence

Deleted.

Line 34 to 35: It's necessary to include this sentence?

Revised sentence to clarify

Line 35 to 39: The results should be described with values ​​and in a clearer way. Here they are written in a broad way and do not attract the reader's attention. Redoing is necessary.

We revised one of these sentences and added values which should improve clarity.  The other sentences describe results with proprietary indices.  The values of these indices have no units and are all relative.  Providing values would not be useful, especially in an abstract where we do not have sufficient space to explain the proprietary indices.  Correspondingly, we describe the relationships of weather metric and proprietary bolus indices. We revised these two sentences to clarify.

Introduction: The objective of this case study was to evaluate the... or The objective of this research was to evaluate? It´s diference  in abstract and introduction. 

Changed “research” to “case study” in the abstract so that they are consistent.

Line 91 to 92: It's 7 to September 4, 2023 and in abstract June 1 to August 29, 2023. Wich?

Lines 91 to 92 refer to the time heifers were kept in the study pasture.  This is provided to let the reader know that heifers were familiar with the pasture.  The actual dates that were evaluated were June 1 to August 29 see the Abstract and line 146-187 in the Methods (highlighted). We used these 3 months to better represent the summer period.

Line 130:  "Precipitation data were not included in the analyses because rainfall occurred periodically with most days having no measurable rainfall. " So precipitation doesn't interfere with the animal's confort?

We could not use precipitation data in our statistical analyses (regression analyses with repeated measures) because it occurred too infrequently.  We added a sentence explaining this on lines 131 and 132 (highlighted).   Instead, we used relative humidity which increases greatly during period of rain in this semi-arid environment. We explain this in the Discussion.

Table 1: Is it necessary to insert this table in the text? This is data that you used for statistics. Describe in the text the average values ​​of each variable within the period studied, with standard deviation.

We used six weather metrics, which describing in the text would be much more cumbersome and harder to read than a table. Tables are easier to read.  We used the minimum and maximum values (in addition to mean) rather than standard deviations of the weather, because it better explains the range of weather that occurred in the study.  It is the extremes (maximums) in weather  that are likely to impact cattle behavior.  Also, this shows the range of weather metrics that help with our visual representations of the relationships we measure.  Correspondingly, we think it is more useful to use minimum, maximum and mean weather values observed in the study.

Line 154 to 175: How many people observed the behaviors? Did you use any reference methodology? Were the observers trained? Describe this information.

There was one observer, and he was trained prior to the study. This was added to the Methodology here (highlighted). References for the definitions of behavior are now provided (highlighted).  Methodology is explained in detail, because the extensive nature of the study pasture and rangeland cattle required modifications of observation methodologies used for collecting training data for supervised machine learning by other researchers in smaller pastures.

Table 2: Is it necessary to insert this table in the text?

Yes, few researchers have used 3-hour intervals for evaluating diurnal patterns in behavior.  This approach (3 hours) provides periods the correspond to typical grazing bouts (e.g., Periods 2 and 6) during the summer.  Seeing the periods should help readers understand our approach and analyses. Tables are much easier to read, and this one should not take up much space in the paper.

Line 499 to 501: And why weren't they recorded at shorter intervals?

The reporting interval for the bolus is determined by the manufacturer.  It is likely set at 10-minute intervals rather than shorter intervals (e.g., 1 minute) to minimize the size of the data packet transferred from the bolus to the reader. More frequent reporting would require larger data packets to be transmitted during the short time periods cattle are near the reader and also increase battery usage. This has been added to the Discussion (highlighted).

Line 599 to 601: This sentence is not necessary.

Sentence deleted.

Reviewer 3 Report

Comments and Suggestions for Authors

The researches are valuable, also very helpful for the field practinioners today and especially in the near future (PLF). The amount of the experimental data is impressive.

However, there are some corrections to be made in order to improve the impact of the article:

- I think (even in the title) that you can use the term welfare instead of well-being. And you have to add Precision Livestock Farming or Precision Livestock Management amont the key-words;

- The most serious problem I was concerned is that you use weather data (RH, AT, wind speed, solar load) recorded by PRA at 6.6 km distance from the research site (according to 2.4 on page 4/25 in Materials and Methods). This likely generates differences, maybe significant. But when I continued to read, I noticed that on the next paragraph, and also in Results chapter (on page 9/25, the phrase before Table 6), you specified that you recorded weather/environment metrics by using Kestrel 5400AG cattle heat load trackers. Therefore I think you have to begin 2.4. (Weather data) by describing the methods used for monitoring Thermal indices, then you can mention that the data are completed (and compared to) the recordings of the PRA weather station;

- It seems to me that the use of epoch for denoting  the behavior observation events is quite forced or innapropriate (on page 5). Maybe you can replace it with another term (i.e. one-minute event?);

- There are some typos (e.g., mintutes, sometimes you wrote Klimo Insights and some other times Kilmo insights etc.). Please correct them.

Please, apply the suggested corrections in the attached .pdf file to improve the value and the impact of your scientific paper.

Author Response

Our responses to Reviewer 3 are best viewed in the Word document below, but we have added our responses below.  Many of reviewer 3 suggestions were on an attached pdf file.  We have incorporated almost all of these suggestions into the manuscript.  The suggestions that we did not incorporate are discussed in Word document and below.  We have highlighted in yellow all our changes to the revised manuscript (including all three of the reviewers).

The researches are valuable, also very helpful for the field practinioners today and especially in the near future (PLF). The amount of the experimental data is impressive.

However, there are some corrections to be made in order to improve the impact of the article:

- I think (even in the title) that you can use the term welfare instead of well-being. And you have to add Precision Livestock Farming or Precision Livestock Management amont the key-words;

Change well-being to welfare in the title and throughout the manuscript (highlighted in yellow).  Also added “precision livestock management to the key-words.

- The most serious problem I was concerned is that you use weather data (RH, AT, wind speed, solar load) recorded by PRA at 6.6 km distance from the research site (according to 2.4 on page 4/25 in Materials and Methods). This likely generates differences, maybe significant. But when I continued to read, I noticed that on the next paragraph, and also in Results chapter (on page 9/25, the phrase before Table 6), you specified that you recorded weather/environment metrics by using Kestrel 5400AG cattle heat load trackers. Therefore I think you have to begin 2.4. (Weather data) by describing the methods used for monitoring Thermal indices, then you can mention that the data are completed (and compared to) the recordings of the PRA weather station;

We believe that the accuracy and quality of weather data collected at the Prescott Regional Airport more than makes up the short distance from the study pasture.  The airport is adjacent to the adjacent pasture to the study pasture.  For rangeland livestock studies the 6.6 km is minimal.  The terrain and elevation from the study pasture and airport are very similar. Many rangeland livestock studies use temporary weather stations or satellite imagery to collect some weather parameters.  These temporary weather stations do not collect the detailed data collected at NOAA weather stations located at a controlled airport.  Prescott Airport collected solar radiation and cloud cover which are not available for affordable temporary weather stations. 

We did include your suggestion to mention that we compared the airport weather values to the Kestrel 5400 AG. The weather metrics recorded by the Kestrel and airport were similar. It would have been impractical to use the Kestrel to collect 24-hour data.

- It seems to me that the use of epoch for denoting  the behavior observation events is quite forced or innapropriate (on page 5). Maybe you can replace it with another term (i.e. one-minute event?);

We changed the term “epoch” to “event” as suggested.

- There are some typos (e.g., mintutes, sometimes you wrote Klimo Insights and some other times Kilmo insights etc.). Please correct them.

Corrected.

Please, apply the suggested corrections in the attached .pdf file to improve the value and the impact of your scientific paper.

We have incorporated almost every one of your suggestions on the .pdf file.

Selected responses to comments on the pdf file. We incorporated most of the suggestions.

Line 15.  We measured the cattle behaviors using the accelerometers in the bolus.

Line 31.  AT is a simple acronym for ambient temperature.  Also, actual temperature and ambient temperature have similar meanings.  We would like to keep AT as the acronym for ambient temperature.

Line 38.  This is an explanation of the statistical relationships and does not directly refer to “recorded values”.  We need to leave this unchanged.

Line 42.  We did not add cattle welfare to the key words since it is in the title. We left well-being in the key words to help with searches.

Line 97 and 104.  We included this information so that the reader could get information on this equipment.  Most journals want the location of the manufacturer of equipment listed.

Line 127-128.  MesoWest is not a commercial company.  It is a service provided by University of Utah to download were data.  To adequately describe where we got the data from the Prescott Airport, this website needs to be provided.

Lines 136, 138-139. The use of websites allows readers to learn more about the equipment we used.  Without these links, how could the reader find out more details about the equipment we used.

Table 1.  Sometimes the relative humidity gets very low in central Arizona.  We obtained this relative humidity level (1.96%)  from the NOAA station at the Prescott Airport.  It should be accurate.

Line 171.  We believe that “up position” is a better term than “higher position”.  For example, higher than what?

Line 175.  We believe that “down position” is a better term than “lower position”.  For example, lower than what?

Line 208.  For repeated measures analyses, the subject is a specific term that is needed to describe the statistical model.

Figure 1.  We felt that it would be better to have the y-axis be reticular temperature to differentiate from ambient temperature that is also used frequently.
